# A Simple In-Vivo Method for Evaluation of Antibiofilm and Wound Healing Activity Using Excision Wound Model in Diabetic Swiss Albino Mice

**DOI:** 10.3390/microorganisms11030692

**Published:** 2023-03-08

**Authors:** Mohammed Alrouji, Fahd A. Kuriri, Mohammed Hussein Alqasmi, Hamood AlSudais, Mohammed Alissa, Meshari A. Alsuwat, Mohammed Asad, Babu Joseph, Yasir Almuhanna

**Affiliations:** 1Department of Clinical Laboratory Sciences, College of Applied Medical Sciences, Shaqra University, Shaqra 11961, Saudi Arabia; 2Department of Clinical Laboratory Sciences, College of Applied Medical Sciences, King Saud University, Riyadh 12372, Saudi Arabia; 3Department of Medical Laboratory Sciences, College of Applied Medical Sciences, Prince Sattam Bin Abdulaziz University, Al-Kharj 11942, Saudi Arabia; 4Clinical Laboratory Sciences Department, College of Applied Medical Sciences, Taif University, Al-Taif 21974, Saudi Arabia

**Keywords:** diabetic, mupirocin, MRSA, crystal violet assay, modified Gram stain

## Abstract

The study developed a simple and inexpensive method to induce biofilm formation in-vivo for the evaluation of the antibiofilm activity of pharmacological agents using Swiss albino mice. Animals were made diabetic using streptozocin and nicotinamide. A cover slip containing preformed biofilm along with MRSA culture was introduced into the excision wound in these animals. The method was effective in developing biofilm on the coverslip after 24 h incubation in MRSA broth which was confirmed by microscopic examination and a crystal violet assay. Application of preformed biofilm along with microbial culture induced a profound infection with biofilm formation on excision wounds in 72 h. This was confirmed by macroscopic, histological, and bacterial load determination. Mupirocin, a known antibacterial agent effective against MRSA was used to demonstrate antibiofilm activity. Mupirocin was able to completely heal the excised wounds in 19 to 21 days while in the base-treated group, healing took place between 30 and 35 days. The method described is robust and can be reproduced easily without the use of transgenic animals and sophisticated methods such as confocal microscopy.

## 1. Introduction

Biofilms are a discrete state of microbial infection, where the microorganisms are enclosed in an extracellular matrix produced by them that is self-assembling and self-supporting [1]. Several acute and chronic infections are known to induce biofilm formation and delay wound healing. Biofilm is reported to impair epithelization and formation of granulation tissue along with inflammation to interfere with wound healing [2,3].

Several in-vitro and in-vivo methods have been reported for biofilm induction and for the evaluation of antibiofilm activities [4]. The in-vitro methods involve biofilm formation in abiotic and biotic surfaces [5]. Compared to the in-vitro methods, the in-vivo methods are more acceptable as it involves the interaction between the host and the microbes and is clinically significant [3]. Several models involving pigs, rats, and mice have been described earlier, each having its own advantages and limitations [3]. Many of these methods involve modern techniques that are expensive as they use different types of microscopes and transgenic animals that are difficult to perform and replicate. Furthermore, some of these methods may not induce biofilm formation easily. Moreover, many of these methods have been conducted using *P. aeruginosa*, poly microbes, and not with other microbes [6,7,8].

Methicillin-resistant *Staphylococcus aureus* (MRSA) is a known pathogen that is responsible for skin and soft tissue infections. Antibiotic resistance coupled with its ability to form biofilms, on both biotic and abiotic surfaces, makes it challenging to treat MRSA infections [9]. There are very few in-vivo biofilm models that have been introduced using MRSA [10,11]. It is one of four bacteria causing wound infection and its treatment is considered a therapeutic challenge. Newer therapeutic strategies can be applied for the treatment of MRSA biofilm if novel and simple methods are available for the evaluation of antibiofilm activity [12]. The present study was undertaken to develop a method to induce and quantify biofilm formation in excision wounds using simple techniques in diabetic mice. The efficacy of antibiotics to treat this biofilm was also evaluated using mupirocin.

## 2. Materials and Methods

### 2.1. Materials

Pressed polyvinyl chloride clear, non-curling plastic coverslips, 0.17 to 0.25 mm thick, 22 mm width (Fisher Scientific, UK) were cut into 10 mm^2^ width and sterilized in UV light were used to induce biofilm on the wound. The bacterial pathogen; Methicillin-resistant *Staphylococcus aureus* (ATCC 43300) available in the department was used in the study. Mupirocin (2% *w*/*w*) ointment was from Avabon Ointment, Avalon Pharma, Riyadh, KSA, and Luria Bertani broth was purchased from SRL Pvt. Ltd., Mumbai, India. Stains and other chemicals were purchased from different chemical suppliers and were of analytical grade. A microscope (Leica DM 2500 LED connected to a DFC 295 camera) was used to capture images.

### 2.2. Animals

Male adult Swiss albino mice weighing between 25 g and 28 g were used. They were kept under a controlled environment and were provided with water and feed *ad libitum*. The methodology was reviewed and approved by the Ethical Research Committee of Shaqra University (No. ERC SU_20220066). All precautions were exercised to prevent the transmission of bacterial pathogens from infected animals to other animals by keeping them in a separate room. Persons handling the animals exercised all the precautions to avoid infection.

### 2.3. In-Vitro Biofilm Formation

This was carried out to determine the biofilm formation ability of the selected MRSA strain. The overnight incubated bacterial culture in a shaking incubator was taken and the cell density was adjusted to 3 × 10^6^ CFU/mL in Luria Bertani (LB) broth. Broth culture (100 μL) was transferred into microtiter plate wells and incubated for 24 h at 37 °C. Biofilm formation was confirmed by the crystal violet binding assay [13]. Briefly, planktonic cells were removed using a micropipette from the corner and the wells were rinsed carefully with sterile distilled water three times successively. Filtered (0.44 µm filter) 0.1% *w/v* crystal violet (20 μL) was added into each well and stained for 10 min. The wells were again rinsed with 10 mM potassium phosphate buffer and air-dried for 15 min. The dye in the wells was solubilized by the addition of 96% *v/v* ethanol (100 µL). After 15 min, the contents were mixed well and the absorbance was measured at 570 nm using the Micro ELISA auto reader (800-TS, Biotek, Winooski, VT, USA).

### 2.4. Preparation of Biofilm on the Coverslips and Its Determination

The MRSA was inoculated into sterilized LB broth in a culture bottle and incubated at 37 °C. Overnight grown cultures were diluted to 0.01 absorbance at OD 600 nm [14]. From this, 5 mL was transferred into fresh LB broth and incubated at 37 °C in a shaker incubator at 210 ± 10 rpm for 3 h. The mid-log phase bacterial cultures were separately transferred into sterile culture tubes. UV light sterilized polyvinyl coverslips were placed in each tube fully immersed in culture media to attain a 90° angle in the tubes relative to the bottom of the tube [15]. Tubes were incubated at 37 °C for 24 h. The coverslips were removed carefully and rinsed with 10 mM potassium phosphate buffer to remove the planktonic cells.

Some of these coverslips were stained for the confirmation and quantification of biofilm formation. The biofilm formation was confirmed by staining with crystal violet for 5 min followed by washing with sterile water. The coverslips were air-dried and observed under a microscope. The quantification of biofilms was carried out by solubilizing the dye with 96% *v/v* ethanol for 20 min and measuring the absorbance at 570 nm.

### 2.5. Induction of Type-II Diabetes in Mice

Type II diabetes was induced by the administration of a combination of streptozocin and nicotinamide [16]. Briefly, animals were fasted for 12 h followed by an injection of nicotinamide (240 mg/kg, i.p). Streptozocin (100 mg/kg, i.p) was administered after 15 min. Seventy-two hours after streptozocin administration, animals were fasted for 12 h and the fasting blood glucose levels were determined. Animals with a fasting blood glucose of more than 150 mg/dL were considered diabetic and these were selected for the study. There was no mortality in animals receiving streptozocin.

### 2.6. Excision Wound and Biofilm Formation

An area of skin was selected in the dorsal region of the mouse and it was depilated using a hair-removing cream (Veet, Reckitt Benckiser Group PLC, Saudi Arabia). The depilated area was thoroughly cleaned using normal saline and the animals were observed for 12 h for any irritation or inflammatory effect of the hair removal cream. On the next day, mice were anesthetized using ketamine (87.5 mg) and xylazine (12.5 mg) cocktail in saline through the intraperitoneal route at a dose of 0.1 mL/20 g [17]. The depilated area of the skin was carefully excised to full thickness and the blood oozing out was cleaned using sterile absorbent cotton. A broth culture (100 µL) containing 10^6^ CFU/mL of MRSA was added to the wound and a coverslip containing the MRSA biofilm was carefully placed over the wounded area. It was held in place using surgical adhesive tape (Smith and Nephew Medical Limited, England). No treatment was given to the animals for the next 72 h. After this period, the adhesive tape along with the coverslip was carefully removed using surgical forceps. Photographs were taken to confirm the infection and biofilm formation on the excised wounds. A thin layer was formed over the wound. Some of these animals were sacrificed by an overdose of ketamine and xylazine cocktail (five times the anesthetic dose). The thin film formed over the wound was removed and part of it was used for bacterial load determination. The remaining part was fixed in neutral formalin (10% *v*/*v*) and subjected to histopathological examination by modified Gram staining [18].

The study was conducted using two groups of animals consisting of 12 animals in each group; the first group animals were diabetic animals induced with biofilm and treated with inert ointment base [19] while the second group animals were diabetic animals induced with biofilm and treated with mupirocin ointment (2% *w*/*w*) [20]. The treatment was applied once daily to completely cover the wounds. The wound area was measured every 4 days for 20 days. At the end of the 20 days, six animals were sacrificed and the tissues were used for histopathological examination and CFU determination. The remaining six animals were treated till the day of epithelization, which is the day of falling of scar without any sign of a wound.

#### Staining

The H and E staining was conducted using standard procedure [21]. The modified Gram’s staining was performed as follows [18]. The paraffin blocks were deparaffinized and hydrated with distilled water. The slides were placed on a staining rack and a drop of crystal violet stain was added onto the tissue section for 1 min followed by washing in tap water. The section was flooded with Lugol’s iodine for 1 min followed by washing with tap water. The sections were blotted dry and acetone was poured over the section until no color came out, followed by washing. The slides were then stained with basic fuchsin for 3 min followed by washing and drying. The slide was quickly dipped into acetone (two dips) and then directly into the picric acid-acetone mixture until a ‘salmon’ color developed. The slide was dipped quickly into two changes of acetone, air-dried, and dipped into xylene. Slides were dried and observed under the microscope attached to a camera (Leica DM 2500 LED connected to a DFC 295 camera).

A diagrammatic representation of the methodology used in the study is given in Figure 1.

## 3. Results

### 3.1. In-Vitro Biofilm Formation

The selected strain of MRSA (ATCC 43300) was able to develop biofilms in-vitro in microtiter plate wells. Measurement of optical density revealed that the MRSA strain used formed a strong biofilm and maximum biofilm was observed at 24 h (Figure 2).

### 3.2. Biofilm Formation in Coverslip and Its Determination

The presence of a confluent stain on the surface of the coverslip indicated the presence of biofilm. Further, observation of the coverslip under 1000× revealed robust biofilm formation with an extracellular matrix (Figure 3). The coverslip crystal violet assay showed that bacterial strain formed maximum biofilm in 24 h and further incubation of the coverslip in the broth did not increase the biofilm formation.

### 3.3. Induction of Type-II Diabetes in Mice

Administration of streptozocin and nicotinamide was effective in inducing diabetes in all the animals. The dose of streptozocin was titrated such that fasting blood glucose levels were between 150 mg/dL and 200 mg/dL in order to avoid mortality due to severe hyperglycemia. The blood glucose levels rose slightly during the experimental period (Figure 4).

### 3.4. Excision Wound in Diabetic Animals

Application of coverslip and broth culture developed a good biofilm over the wounds in 72 h. The wound tissue appeared grayish with pus formation and exudation. Determination of the bacterial load of the wounded tissue at 72 h showed that it was between 5.32 and 6.38 log CFU/g of wounded tissue. Histological examination of wounded tissue after 72 h showed co-aggregates and flocculates of bacterial cells confirming the biofilm formation (Figure 5).

The wound contraction as an index of wound healing was measured every 4 days for 20 days after biofilm formation. The day of biofilm formation was considered as ‘day 0’. Topical application of antibiotic ointment significantly increased the healing of the wounds compared to the ointment base-treated group. The bacterial load (CFU/g tissue) in antibiotic-treated animals was significantly less compared to the base-treated group indicating that infection can be healed by antibiotic application (Figure 6). The wounds were partially healed in the base-treated group on day 20 while the antibiotic-treated group showed good wound healing (Figure 7). Histological examination of the tissue stained with H and E supported the macroscopic observation on wound healing as the epidermal regeneration was noticeably higher in the antibiotic-treated group. Modified Gram staining of the tissue on the 20th day showed fewer bacterial cells in the antibiotic-treated group compared to the base-treated group (Figure 8). The epithelization period, which is the day on which there is no wound, was between 19 and 21 days for the antibiotic-treated group and between 30 and 35 days for the base-treated group.

## 4. Discussion

A simple method with the use of easily available instruments and chemicals to induce biofilm has been outlined in this study. The method used cover slips, a compound microscope, and simple media along with basic staining techniques. The method described is inexpensive and uncomplicated; it is easy to perform in basic laboratory settings.

In the current study, we utilized mice to produce biofilm keeping in mind the 3Rs for experimentation that recommends the use of the smallest animal possible [22]. Furthermore, many of the earlier reported methods used transgenic animals, and hairless (nude) animals with sophisticated techniques such as the use of confocal microscopy to confirm the biofilm formation [4]. In both the confocal microscopy and compound microscopy that was used in the present study, the biofilm developed was confirmed by characteristics such as an extracellular matrix. The extracellular matrix was clearly visible under the compound microscope. Confocal microscopy can be used for both qualitative and quantitative measurements of extracellular matrix [23] while the limitation of compound microscopy is that it can be used for qualitative observation. This limitation can be overcome using image analysis software.

In the current study, diabetic mice were used for the development of biofilm on the skin wound. It is difficult to form biofilm using normal animals. Diabetes created a favorable environment through the synergistic action of the pathogen to induce biofilm [24]. Furthermore, there are earlier reports on biofilm formation in diabetic animals using *P. aeruginosa* [25]. Several methods are available for inducing diabetes in animals. In the present study, type-II diabetes was induced using a combination of nicotinamide and streptozocin. In this method, the insulin-secreting β-cell of pancreatic islets are not damaged completely by the streptozocin due to the protective effect of nicotinamide [26]. Injection of streptozocin only should be avoided as it leads to the development of type-I diabetes through the autoimmune damage of pancreatic β-cells [27]. The type-II diabetes model is better than a type-I diabetes model because wound infection along with severe hyperglycemia may lead to more mortality among the animals unless these animals are injected with insulin to maintain their blood glucose levels. Furthermore, the administration of insulin may influence the normal healing of wounds due to its cell-proliferating effects [28]. A number of skin infection models such as abrasion, superficial incision with foreign body, blade scrape, and burn infections along with excision wound models have been described earlier by different authors [29]. In the incision wound model, burn wound model, sanding model, and biopsy punch, it is difficult to isolate the biofilm formed over the wounded area. In the excision wound, the whole skin is excised thereby exposing the tissues beneath the skin and the formed biofilm can be easily seen by the naked eye as a grayish film. Additionally, in our study, we removed this grayish film to confirm microscopically the formation of biofilm. Moreover, induction and quantification of wound healing are much easier in the excision wound model compared to other wound healing models. Wound healing can be determined easily by measuring the wounded area whose periphery is clearly noticeable on daily basis.

Furthermore, the excision wound models described earlier in the mice used either an inoculation of bacterial culture or the application of preformed biofilm on the wound [4]. We tried to induce biofilm formation using one of these methods by inoculating an MRSA culture on the excised skin in diabetic animals reported by Zhao et al. [30], but the biofilm formation was poor and there was no film formation in some of the animals. The exact reason why the above-reported method could not be reproduced is not known. However, we believe it could be due to the difference in the strain of the animal used. Zhao et al. [30] used transgenic diabetic mice (db/db) for biofilm formation while we tried to induce the biofilm using commonly available Swiss albino mice. The db/db mouse is a genetically mutated mouse that is extremely obese having metabolic disorders with a defect in leptin function [31]. Furthermore, the above method used *P. aeruginosa*, which is reported to produce biofilms more easily compared to MRSA [32]. Hence, the method used in the present study is inexpensive because it does not use transgenic animals and may be more reproducible with other strains of bacteria as MRSA does not produce biofilm as easily as *P. aeruginosa*.

Another method described to induce biofilm by Agostinho et al. [33] used preformed *P. aeruginosa* biofilm in SKH-1 hairless diabetic mice. In this method, the biofilm was formed on a transparent sheet and it was applied to the excised skin of the hairless mouse. Diabetes was induced by a single dose of streptozocin. Again, we tried to replicate this method in Swiss albino mice using a single dose of streptozocin (type-I diabetes) as well as a combination of streptozocin and nicotinamide (type-II diabetes) [34]. The biofilm formation was poor in both cases indicating that it may be due to the difference in the strain of the animals and the bacteria used. Therefore, the method described by us is more robust and inexpensive that can be reproduced easily.

The management of biofilms in wounds is a concern in clinical settings and a number of antimicrobials are used to control these infections. Many of these conventionally used antimicrobials are ineffective due to the development of microbial resistance [35]. Agents that eradicate biofilm are useful in preventing infection-related complications [36,37]. Both modern drugs and plant-derived products are being evaluated for anti-biofilm effects. The method described by us can be used for the evaluation of pure compounds as well as crude plant extracts for anti-biofilm effects. The plant extracts, phytoconstituents, or pure compounds must be suitably tested in different ointment bases that should include both hydrophilic and hydrophobic bases for the release of active constituents before their application onto the wounds. This method may not be suitable for the evaluation of orally administered compounds or plant-derived products as handling the animals for oral administration may affect the biofilm and the wound healing activity.

There are a few important points that have to be considered while using the method described in the current study. First, is the method for the induction of diabetes in mice. The injection of a single dose of streptozocin, which is commonly used to induce type-I diabetes may not bear a similar result. The reason is that a single injection of streptozocin causes the slow destruction of beta-cells leading to severe diabetes with time [38]. This may increase the mortality as the animals suffer from profound hyperglycemia with severe infection. Another precaution to be exercised is blood glucose levels. We titrated the dose of streptozocin to produce mild diabetes in the animals. Animals with fasting glucose levels between 150 mg/dL and 200 mg/dL were selected for the study. Some animals that were severely diabetic with blood glucose levels of more than 200 mg/dL at the beginning of the experiment succumbed to the infection within 5 to 7 days after inoculation of bacterial culture.

The second point to be considered is the method for the development of biofilm on the excised wound. The application of a preformed biofilm coverslip without the addition of 100 µL of bacterial culture does not induce biofilm formation. The third and last is the confirmation of biofilm 72 h after inoculation of the bacterial culture by histological studies. Macroscopic features are not sufficient to confirm the biofilm formation on the wound. Histological studies using modified Gram staining along with bacterial load determination should be carried out to confirm the biofilm formation on the wound.

Factors that affect biofilm formation such as surface adhesion, time limit, the capacity of the pathogen to form a biofilm, environmental stress, and duration of healing should be considered while using the excision wound model in mice.

## 5. Conclusions

The paper described an inexpensive method for the induction of biofilm in the wounds of commonly used Swiss albino mice. The wounds were induced using an excision wound model and the animals were made diabetic using streptozocin and nicotinamide. Biofilm formation was conducted using coverslips along with the inoculation of a bacterial culture. Macroscopic, microscopic, and bacterial load in the wounded tissue indicate that the method developed is reproducible and can be used for the evaluation of pharmacological agents for antibiofilm activity.

## Figures and Tables

**Figure 1 microorganisms-11-00692-f001:**
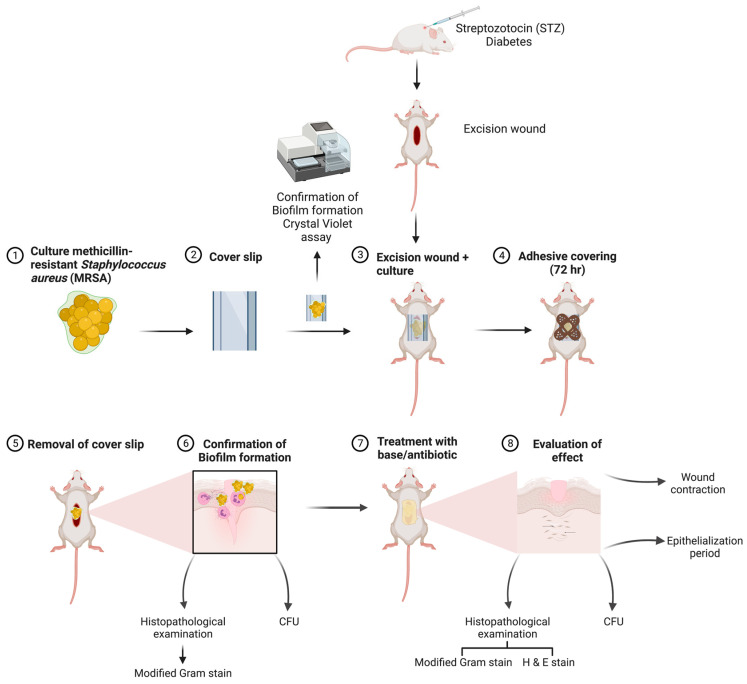
Diagrammatic representation of the methodology.

**Figure 2 microorganisms-11-00692-f002:**
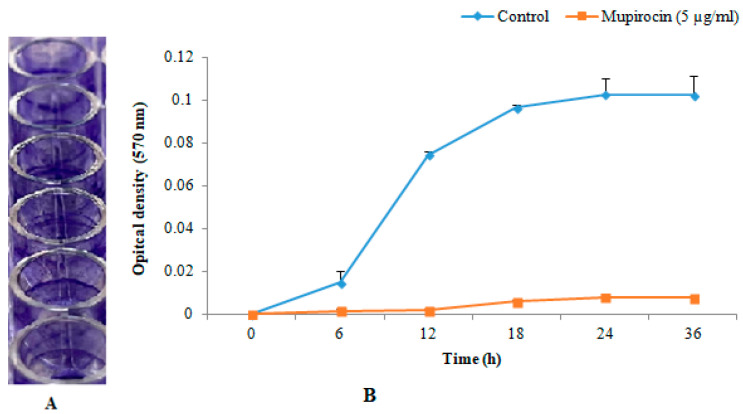
In-vitro biofilm formation. Biofilm formation on microtiter plate well (**A**). Graph showing optical density in crystal violet assay (**B**).

**Figure 3 microorganisms-11-00692-f003:**
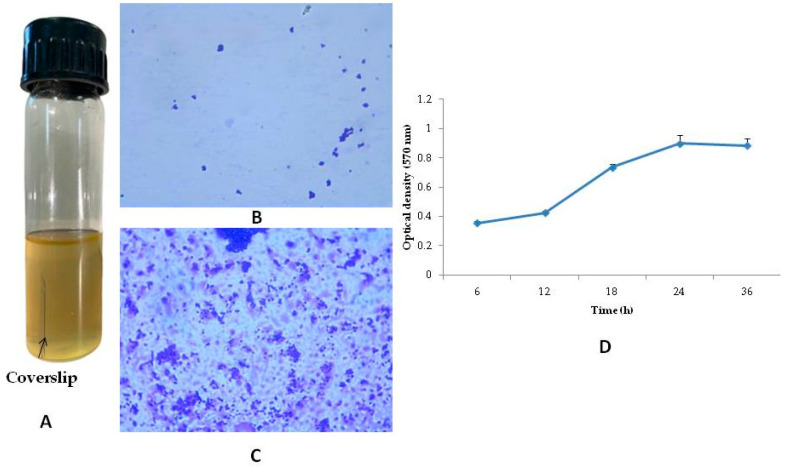
Biofilm formation setup in the coverslip. Coverslip in the media (**A**), microscopic image (1000×) of MRSA in broth (**B**), microscopic image (1000×) of biofilm formation on the coverslip (**C**), measurement of optical density in coverslip crystal violet assay at 570 nm (**D**).

**Figure 4 microorganisms-11-00692-f004:**
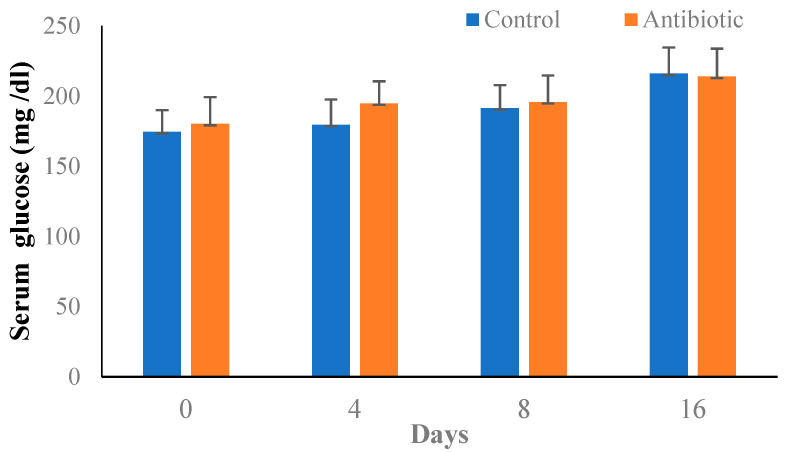
Blood glucose levels on different days during the experimental period in mice.

**Figure 5 microorganisms-11-00692-f005:**
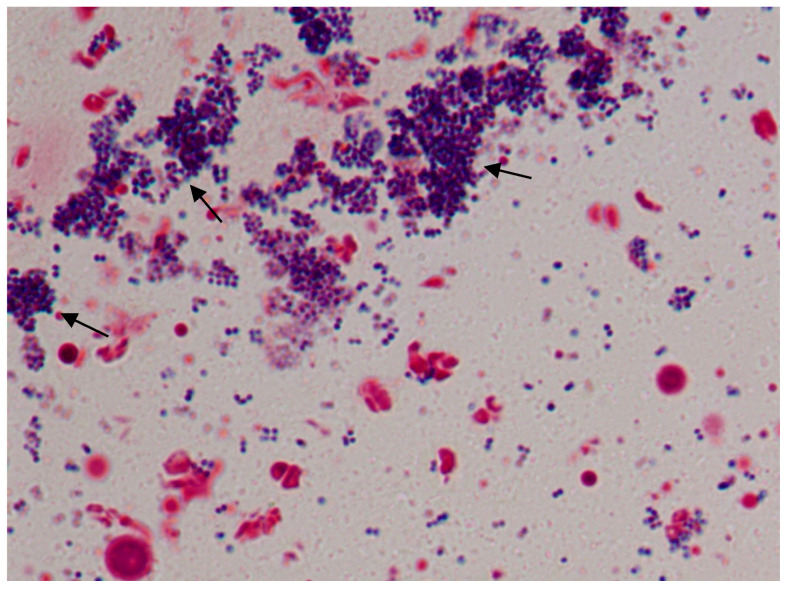
Biofilm formation on the wounded skin 72 h after inoculation (Modified Gram stain—1000×).

**Figure 6 microorganisms-11-00692-f006:**
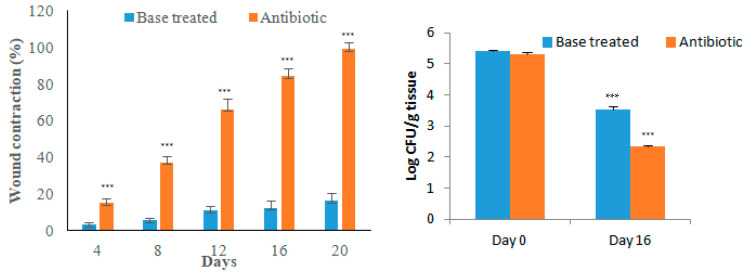
Healing of excision wound in MRSA-induced biofilm. The left side of the figure shows wound contraction on different days and the right side shows bacterial load in the wounded tissue. All values are mean ± SEM, n = 6, *** *p* < 0.001.

**Figure 7 microorganisms-11-00692-f007:**
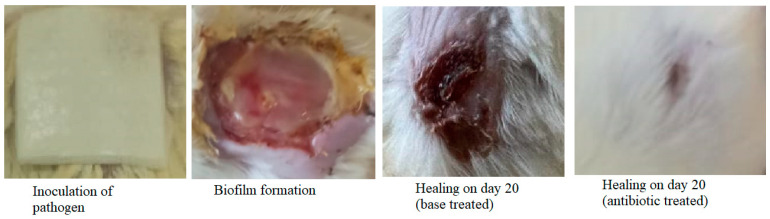
Photographs showing inoculation of the pathogen, biofilm formation, and healing on day 20 in base-treated and antibiotic-treated groups.

**Figure 8 microorganisms-11-00692-f008:**
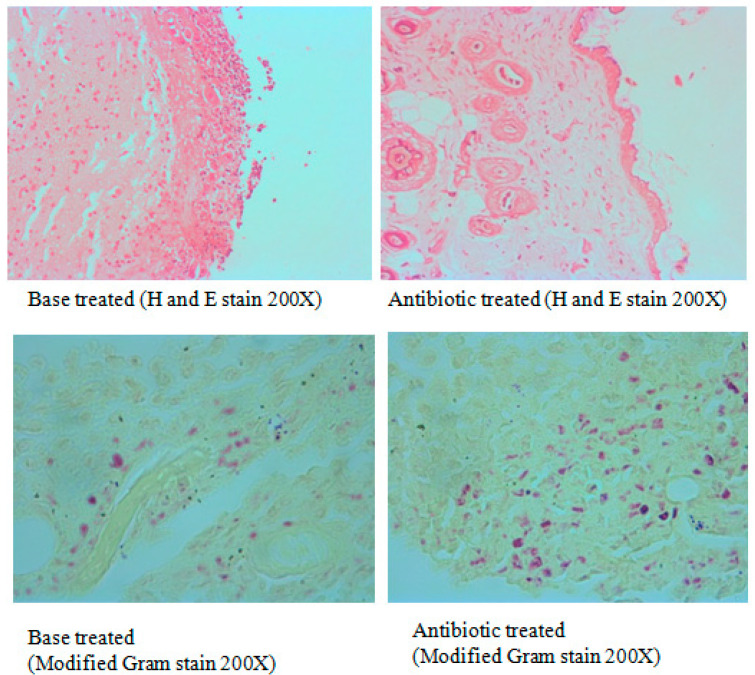
Photomicrographs showing healing of excision wound in the base-treated and antibiotic-treated group on day 20.

## Data Availability

Raw data will be provided on request.

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
