# Peer review of "A Simple In-Vivo Method for Evaluation of Antibiofilm and Wound Healing Activity Using Excision Wound Model in Diabetic Swiss Albino Mice"

_microorganisms, 2023, doi:10.3390/microorganisms11030692_

Round 1
Reviewer 1 Report
The article "A simple in-vivo method for evaluation of antibiofilm and
wound healing activity using excision wound model in diabetic
Swiss albino mice is devoted to the development of a mouse model for studying the healing of wounds infected with biofilms." The manuscript relates to topical issues and is of interest to readers of Microorganisms, however, some aspects of the study and the proposed model turned out to be unclear.
Major points:
1. Comparison of the proposed method with known animal models, although given in the introduction, needs to be substantially deepened. In particular, the need to create diabetes in mice is not explained, is this related to the need to study chronic wounds?
2. Existing in vivo models differ in the mechanisms by which wounds are inflicted. In the manuscript authors use excision wounds rather than incision and burn wounds, sanding wounds or biopsy punch. Alternative approaches to the wound formation are mentioned in the discussion, but there is no clear justification for the choice of excision wounds.
3. Authors claim that the model does not require use of confocal microscopy. Nonetheless, there is no validation of the model with confocal microscopy, so one can not be sure that the proposed model provides the same results as controlled with confocal microscopy.
Author Response
Reviewer #1
- Comparison of the proposed method with known animal models, although given in the introduction, needs to be substantially deepened. In particular, the need to create diabetes in mice is not explained, is this related to the need to study chronic wounds?
We appreciate the comments of the reviewer. A detailed explanation of the reasons for choosing diabetic animals for the development of biofilm is now given under the discussion.
“In the current study, diabetic mice were used for the development of biofilm on the skin wound. It is difficult to form biofilm using normal animals. Diabetes created a favorable environment through the synergistic action of the pathogen to induce biofilm [24]. Furthermore, there are earlier reports on biofilm formation in diabetic animals using P. aeruginosa [25]. Several methods are available for inducing diabetes in animals. In the present study, type-II diabetes was induced using a combination of nicotinamide and streptozocin. In this method, the insulin-secreting b-cell of pancreatic islets are not damaged completely by the streptozocin due to the protective effect of nicotinamide [26]. Injection of streptozocin only should be avoided as it leads to the development of type-I diabetes through autoimmune damage of pancreatic b-cells [27]. Type-II diabetes model is better than a type-I diabetes model because wound infection along with severe hyperglycemia may lead to more mortality among the animals unless these animals are injected with insulin to maintain their blood glucose levels. Furthermore, the administration of insulin may influence the normal healing of wounds due to its cell-proliferating effects [28].”.
- Existing in vivo models differ in the mechanisms by which wounds are inflicted. In the manuscript authors use excision wounds rather than incision and burn wounds, sanding wounds or biopsy punch. Alternative approaches to the wound formation are mentioned in the discussion, but there is no clear justification for the choice of excision wounds.
We agree with the reviewer that there are other models to study wound infection. “In the incision wound model, burn wound model, sanding model, and biopsy punch, it is difficult to isolate the biofilm formed over the wounded area. In the excision wound, the whole skin is excised thereby exposing the tissues beneath the skin and the formed biofilm can be easily seen by the naked eye as a grayish film. Additionally, in our study, we removed this grayish film to confirm microscopically the formation of biofilm. Moreover, induction and quantification of wound healing is much easier in the excision wound model compared to other wound healing models. Wound healing can be determined easily by measuring the wounded area whose periphery is clearly noticeable on daily basis”. This is now included in the discussion.
- Authors claim that the model does not require use of confocal microscopy. Nonetheless, there is no validation of the model with confocal microscopy, so one can not be sure that the proposed model provides the same results as controlled with confocal microscopy.
Confocal microscopy is not a model for biofilm formation, but it is one of the methods used to confirm the development of biofilm. “In both confocal microscopy and compound microscopy that was used in the present study, the biofilm developed is confirmed by characteristics such as an extracellular matrix. The extracellular matrix was clearly visible under the compound microscope. Confocal microscopy can be used for both qualitative and quantitative measurements of extracellular matrix [23] while the limitation of compound microscopy is that it can be used for qualitative observation. This limitation can be overcome using image analysis software”. This limitation is now mentioned in the discussion.

Reviewer 2 Report
The manuscript entitled "A simple in-vivo method for evaluation of antibiofilm and wound healing activity using excision wound model in diabetic Swiss albino mice" is written as per the Journal style. however, it requires a few minor corrections which are mentioned in the as highlighted and commented in the pdf file.

Author Response
Reviewer # 2
The manuscript entitled "A simple in-vivo method for evaluation of antibiofilm and wound healing activity using excision wound model in diabetic Swiss albino mice" is written as per the Journal style. however, it requires a few minor corrections which are mentioned in the as highlighted and commented in the pdf file.
All corrections shown by the reviewer have been incorporated.
